# Lipocalin-2 Deficiency Diminishes Canonical NLRP3 Inflammasome Formation and IL-1β Production in the Subacute Phase of Spinal Cord Injury

**DOI:** 10.3390/ijms24108689

**Published:** 2023-05-12

**Authors:** Nina Müller, Miriam Scheld, Clara Voelz, Natalie Gasterich, Weiyi Zhao, Victoria Behrens, Ralf Weiskirchen, Maryam Baazm, Tim Clarner, Cordian Beyer, Nima Sanadgol, Adib Zendedel

**Affiliations:** 1Institute of Neuroanatomy, RWTH University Hospital Aachen, 52074 Aachen, Germany; 2Institute of Molecular Pathobiochemistry, Experimental Gene Therapy and Clinical Chemistry (IFMPEGKC), RWTH University Hospital Aachen, 52074 Aachen, Germany; 3Department of Anatomy, School of Medicine, Arak University of Medical Sciences, Arak 38481-7-6341, Iran; 4Institute of Anatomy, Rostock University Medical Center, 18057 Rostock, Germany; 5Institute of Anatomy, Department of Biomedicine, University of Basel, 4001 Basel, Switzerland

**Keywords:** NLRP3, inflammasome, spinal cord, pyroptosis, neuroinflammation

## Abstract

Spinal cord injury (SCI) results in the production of proinflammatory cytokines due to inflammasome activation. Lipocalin 2 (LCN2) is a small secretory glycoprotein upregulated by toll-like receptor (TLR) signaling in various cells and tissues. LCN2 secretion is induced by infection, injury, and metabolic disorders. In contrast, LCN2 has been implicated as an anti-inflammatory regulator. However, the role of LCN2 in inflammasome activation during SCI remains unknown. This study examined the role of *Lcn2* deficiency in the NLRP3 inflammasome-dependent neuroinflammation in SCI. *Lcn2^−/−^* and wild-type (WT) mice were subjected to SCI, and locomotor function, formation of the inflammasome complex, and neuroinflammation were assessed. Our findings demonstrated that significant activation of the HMGB1/PYCARD/caspase-1 inflammatory axis was accompanied by the overexpression of LCN2 7 days after SCI in WT mice. This signal transduction results in the cleaving of the pyroptosis-inducing protein gasdermin D (GSDMD) and the maturation of the proinflammatory cytokine IL-1β. Furthermore, *Lcn2^−/−^* mice showed considerable downregulation in the HMGB1/NLRP3/PYCARD/caspase-1 axis, IL-1β production, pore formation, and improved locomotor function compared with WT. Our data suggest that LCN2 may play a role as a putative molecule for the induction of inflammasome-related neuroinflammation in SCI.

## 1. Introduction

Spinal cord injury (SCI) often results from an abrupt traumatic impact on the spine, leading to progressive central nervous system (CNS) degeneration and cystic cavitation within the spinal cord that develops away from the initial trauma site. The pathological processes of SCI are dynamic and divided into three main phases: (i) a primary phase resulting from the initial injury, which is characterized by the destruction of nerve tissue and blood vessels; (ii) a secondary phase, which follows the initial primary injury and is driven by various pathophysiological mechanisms, including neuroinflammation, neurotransmitter-induced excitotoxicity, and oxidative stress; and finally, (iii) a chronic phase, which is specified by extensive scar formation and the progression of cystic cavities [1].

Neuroinflammation plays a crucial pathological role in SCI, and inflammatory cytokines such as IL-1β have been identified as critical cytokines in the etiology of SCI at the cellular level. The activation of inflammasomes and their associated components is responsible for the maturation and secretion of IL-1β [2]. The inflammasome response involves two steps, first, the priming, resulting in the upregulation of its components, followed by the activation, resulting in cytokine release. Triggered by pathogen-associated pattern molecules (PAMPs) and damage-associated pattern molecules (DAMPs) that bind to toll-like receptor 4 (TLR4), the processing of caspase-1 (CASP1) and the secretion of IL-1β are initiated. The NLRP3 inflammasome consists of the sensor protein NOD-like receptor (NLR), an adaptor molecule apoptosis-associated speck-like protein containing a CARD (ASC, PYCARD) and an enzymatic component: CASP1 [3]. We and others have shown that SCI triggers NLRP3 inflammasome activation in the spinal cord following a traumatic injury [4,5,6]. Activation of the inflammasome causes the cleavage of the precursors of IL-1β, IL-18, and gasdermin-D (GSDMD) into their mature forms, which then trigger inflammatory processes and pyroptosis [7]. The GSDMD N-terminal active fragment (GSDMD-NT) has been identified as a downstream molecule of inflammasome signaling. It is typically activated by CASP1 and provokes pyroptosis and the subsequent secretion of mature IL-1β [8,9].

Lipocalin-2 (LCN2), also known as neutrophil gelatinase-associated lipocalin (NGAL), is an antibacterial secreted protein of the lipocalin family involved in the control of various biological processes, such as responding to bacterial infection, cell migration, [10] and innate immunity [11]. Neuroinflammation can promote the secretion of LCN2 from the liver, intestine, glia, and endothelial cells [12]. Consequently, LCN2 plays a crucial role in cell migration and the recruitment of brain-intrinsic and peripheral immune cells to the injury site [13]. In this regard, LCN2 has been shown to contribute to the inflammatory reaction by increasing the expression of inflammatory cytokines and exacerbating the pathological process in some disease models, including stroke [11] and nonalcoholic steatohepatitis [14]. However, LCN2 has also been shown to act as an anti-inflammatory mediator in other CNS diseases, such as experimental autoimmune encephalomyelitis (EAE), a model of multiple sclerosis (MS) [15], and LPS-induced stimulation of bone marrow–derived macrophages [16]. In addition, our previous studies using a mouse model of SCI showed that LCN2 is increased in spinal cord lesions and blood circulation [17].

It is becoming clearer how LCN2 acts in the innate immune system’s protection against pathogens and infections. Its involvement in other inflammatory diseases and severe traumas, however, has not been fully investigated. Numerous associations between LCN2 and the NLRP3 inflammasome complex have been found in earlier studies. For example, it has been shown that the NLRP3 inflammasome attenuates LCN2 secretion in LPS-stimulated macrophages [18]. Another report using a metabolism-associated fatty liver disease model showed that higher systemic LCN2 concentrations trigger NLRP3 activation in brain endothelial cells. The effect appears to be mediated by affecting the release of the high mobility group protein B1 (HMGB1), which can bind and act as a DAMP to the TLR4 receptor [14]. 

We present evidence that LCN2 is expressed after SCI and that LCN2 has a negative impact on SCI by causing the loss of neurons and astrocytes, the release of proinflammatory cytokines, and the invasion of immune cells. In another study, we showed that NLRP3 inflammasome complexes and their modules were successively activated in the zone of injury to tissue following SCI. Characterizing LCN2’s function during neuroinflammatory reactions in the CNS is of critical importance given the crucial roles that LCN2 and NLRP3 play in the pathophysiology of CNS diseases. We thus aimed to determine if LCN2 affects the activation of NLRP3 and inflammasome and acts as an appropriate target to enhance the outcome following SCI in the current work using *Lcn2^−/−^* mice.

## 2. Results

### 2.1. Prediction of the Relationship between LCN2 and NLRP3

The construction of the protein–protein interaction (PPI) network and the identification of the hub genes indicated no direct relationship and interaction between LCN2 and NLRP3. On the other hand, the data showed that these proteins could influence each other via the IKKα/NF-κB pathway and also link proteins such as MMP9 and CASP1 (Figure 1A). Evaluation of the relationship between LCN2 or NLRP3 and human diseases showed that both are associated with a more significant number of genes involved in major inflammatory diseases. Furthermore, besides inflammation, LCN2 was connected to neurodegenerative diseases, especially Alzheimer disease, while NLRP3 was associated with multiple sclerosis (Figure 1B).

### 2.2. SCI Leads to Increased Neuroinflammation in a Time-Dependent Manner

HMGB1 is a proinflammatory mediator that can act as a DAMP by binding to TLR4, thereby promoting the expression of inflammasome-related genes. We assessed the mRNA and protein expression of HMGB1 in wild-type (WT) mice and found a significant induction of the *Hmgb1* mRNA (*p* = 0.0006, *df =* 5, *F =* 5.287) and the HMGB1 protein (*p* = 0.0159, *df =* 5, *F =* 4.305) 7 days post-SCI compared with sham (Figure 2A). Since the processing and maturation of IL-1β and GSDMD requires the activation of CASP1, we analyzed the expression profile of CASP1 during the first 7 days post-SCI (Figure 2B). The mRNA expression of *Casp1* was significantly increased starting at 72 h (*p* = 0.0180, *df =* 5, *F =* 19.93) with a peak at 7 days (*p* < 0.0001, *df =* 5, *F =* 19.93) post-SCI compared with sham (Figure 2B). At the protein level, pro-CASP1 was significantly increased compared with sham starting at 12 h (*p* = 0.0079, *df =* 5, *F =* 10.09) post-SCI and peaking at 7 days (*p* = 0.0001, *df =* 5, *F =* 10.09) post-SCI (Figure 2B). A significant increase in the active form CASP1 was found at 72 h (*p* = 0.0108, *df =* 5, *F =* 8.290) and 7 days (*p* = 0.0002, *df =* 5, *F =* 8.290) post-SCI compared with sham (Figure 2C). Additionally, the mRNA expression of *Gsdmd* was significantly increased starting at 72 h post-SCI (*p* = 0.0034, *df =* 5, *F =* 23.25) with a peak at 7 days (*p* < 0.0001, *df =* 5, *F =* 23.25) post-SCI compared with sham (Figure 2C). Analysis of the corresponding protein by Western blot revealed that the protein levels of the preform increased significantly only at 72 h post-SCI (*p =* 0.0044, *df =* 5, *F =* 6.066), while the active form GSDMD-NT was also increased at 7 days post-SCI (*p* < 0.0001, *df =* 5, *F =* 27.49) compared with sham (Figure 2C). Finally, we analyzed the mRNA and protein levels of the proinflammatory cytokine IL-1β. We observed an upregulation at 12 h (*p =* 0.0001, *df =* 5, *F =* 7.968) and 24 h (*p =* 0.0171, *df =* 5, *F =* 7.968) post-SCI compared with sham at the mRNA level (Figure 2D). Regarding the protein level, a significant increase in pro-IL-1β was found at 72 h (*p =* 0.0108, *df =* 5, *F =* 6.428) and 7 days (*p =* 0.0125, *df =* 5, *F =* 6.428) post-SCI compared with sham (Figure 2D).

### 2.3. SCI Time-Dependently Increased Inflammasome Activation in Neuron and Glia Cells

In the next step, we aimed to determine the expression pattern of the NLRP3 inflammasome complexes. For this purpose, NLRP3 and PYCARD gene and protein levels were analyzed (Figure 3A). The *Nlrp3* mRNA displayed a gradual increase at 12 h (*p =* 0.0157, *df =* 5, *F =* 13.68), 24 h (*p =* 0.0120, *df =* 5, *F =* 13.68), and 72 h (*p =* 0.0014, *df =* 5, *F =* 13.68), where it reached a peak at 7 days (*p* < 0.0001, *df =* 5, *F =* 13.68) post-SCI compared with sham (Figure 3A). Moreover, the NLRP3 protein showed a gradual increase at 72 h (*p =* 0.0239, *df =* 5, *F =* 8.780) and reached a peak at 7 days (*p =* 0.0008, *df =* 5, *F =* 8.780) post-SCI compared with sham (Figure 3A). On the other hand, the *Pycard* mRNA displayed a gradual increase at 72 h (*p =* 0.0009, *df =* 5, *F =* 10.38) and reached a peak at 7 days (*p* < 0.0002, *df =* 5, *F =* 10.38) post-SCI compared with sham (Figure 3A). Moreover, the PYCARD protein showed a significant increase only at 7 days (*p =* 0.0004, *df =* 5, *F =* 8.147) post-SCI compared with sham (Figure 3A).

Colocalization of NLRP3 with neuron (NeuN), astrocyte (GFAP), microglia (AIF1), and oligodendrocyte (OLIG2) markers was investigated by double immunofluorescence staining in the spinal cord after 24 h and 7 days post-SCI (Figure 3B). It revealed that NLRP3 is coexpressed in all studied cell types in a time-dependent manner, increasing following SCI (Figure 3B). After 24 h (*p =* 0.0227, *df =* 2, *F =* 8.309; *p =* 0.0424, *df =* 2, *F =* 16.85) and 7 days (*p =* 0.0382, *df =* 2, *F =* 8.309; *p =* 0.0028, *df =* 2, *F =* 16.85) post-SCI, there was a statistically significant increase in the number of NLRP3^+^NeuN^+^ and NLRP3^+^GFAP^+^ double-positive cells compared with sham (Figure 3B). NLRP3^+^AIF1^+^ and NLRP3^+^OLIG2^+^ double-positive cells were only revealed at 7 days (*p =* 0.0051, *df =* 2, *F =* 13.46; *p =* 0.0029, *df =* 2, *F =* 16.59) post-SCI compared with sham (Figure 3B). Of all cell types, astrocytes showed the most significant increase for NLRP3 protein expression in both time points, 24 h and 7 days post-SCI (Figure 3B). Therefore, astrocytes are the promising glia type for a more detailed NLRP3 inflammasome evaluation.

### 2.4. Lcn2 Deficiency Significantly Improved Locomotor Function after SCI

To investigate the effect of *Lcn2* deficiency on locomotor function after SCI, we included *Lcn2^−/−^* mice in our study. The BBB score of a healthy mouse is 21, which equals the total locomotor activity. In the sham group, the values mainly were at 21, with a slight reduction after the surgical intervention that normalized immediately (Figure 4A). After the injury, the BBB score was 0 in both genotypes, accounting for a successful SCI intervention. Then, it gradually increased with no significant difference between the genotypes until 5 days after the injury (Figure 4A). At days 5 (*p =* 0.0288) and 7 (*p =* 0.0107) post-SCI, *Lcn2^−/−^* mice displayed a significantly higher BBB score compared with the WT animals, indicating a better functional recovery in this group (Figure 4A). We should remind that we observed a significant increase in the LCN2 protein at 7 days post-SCI (*p =* 0.0092, *df =* 1, *F =* 11.18) compared with sham in WT mice (Figure 4B). *Lcn2^−/−^* mice did not show any LCN2 protein expression in sham or 7 days post-SCI (Figure 4B).

### 2.5. Lcn2 Deficiency Significantly Decreased Inflammasome Formation

After the evaluation of the NLRP3 formation in the spinal cord, we observed that the most significant increase was observed 7 days post-SCI (Figure 3A). To investigate the effect of *Lcn2* deficiency on the inflammasome formation 7 days post-SCI, we included *Lcn2^−/−^* mice in our study. We assessed the mRNA and protein expression of HMGB1 in *Lcn2^−/−^* mice and found a significant reduction in the mRNA (*p =* 0.0016, *df =* 1, *F =* 12.74) and protein (*p* < 0.0001, *df =* 1, *F =* 25.00) at 7 days post-SCI compared with WT mice (Figure 4C). The *Nlrp3* mRNA expression at 7 days post-SCI decreased significantly (*p* = 0.0090, *df =* 1, *F =* 8.731) in *Lcn2^−/−^* mice compared with WT mice (Figure 4D). At the protein level, *Lcn2^−/−^* mice displayed a significantly (*p* = 0.0164, *df =* 1, *F =* 5.559) lower expression of NLRP3 7 days post-SCI when compared with WT mice (Figure 4D). Finally, the mRNA (*p =* 0.0186, *df =* 1, *F =* 6.547) and protein (*p =* 0.0002, *df =* 1, *F =* 23.16) levels for PYCARD were lower in *Lcn2^−/−^* mice compared with WT mice 7 days post-SCI (Figure 4D).

### 2.6. Activation of CASP1/GSDMD-NT/IL-1β Axis Reduced in Lcn2^−/−^ Mice

After the evaluation of inflammasome activation in the spinal cord, we observed that the most significant increase was observed at 7 days post-SCI (Figure 2). Therefore, to investigate the effect of *Lcn2* deficiency on inflammasome activation 7 days post-SCI, we included *Lcn2^−/−^* mice in our study. It is confirmed that the activation of the NLRP3 inflammasome results in the activation of CASP1, leading to the cleavage of GSDMD and pro-IL-1β into their active forms GSDMD-NT and IL-1β. To test whether LCN2 influences the activation of CASP1, we analyzed the expression of active CASP1, IL-1β, and GSDMD-NT in WT and knockout mice 7 days post-SCI. The *Casp1* mRNA levels were significantly decreased (*p* = 0.0088, *df* = 1, *F* = 8.497) in *Lcn2^−/−^* mice compared with WT mice 7 days post-SCI (Figure 5A). In accordance, the protein quantities of the pro (*p* = 0.0264, *df* = 1, *F* = 5.182) and active (*p* < 0.0001, *df* = 1, *F* = 28.17) forms of CASP1 in the knockout group decreased and were almost undetectable in *Lcn2^−/−^* mice compared with WT mice (Figure 5A). On the other side, *Lcn2* deficiency had no significant effect (*p* = 0.0947, *df* = 1, *F* = 4.287) on the expression of *Gsdmd* mRNA 7 days post-SCI (Figure 5B). Semiquantitative protein analysis of the full-length GSDMD protein showed only a weak significant decrease (*p* = 0.0499, *df* = 1, *F* = 9.088) in its expression in *Lcn2^−/−^* mice compared with WT mice (Figure 5B). Analysis of protein levels of GSDMD-NT revealed a significant decrease (*p* = 0.0029, *df* = 1, *F* = 11.76) after 7 days post-SCI in *Lcn2^−/−^* mice compared with WT mice (Figure 5B). Finally, *Il-1β* gene expression was decreased 7 days post-SCI in *Lcn2^−/−^* mice (*p* = 0.0003, *df* = 1, *F* = 19.95) compared with WT mice (Figure 5C). On the protein level, both pro (*p* = 0.0250, *df* = 1, *F* = 4.181) and active (*p* = 0.0002, *df* = 1, *F* = 18.52) forms of IL-1β were found to be expressed lower in *Lcn2^−/−^* mice than in WT mice 7 days post-SCI (Figure 5C).

### 2.7. Lcn2^−/−^ Led to Reduced NLRP3 Activation in Astrocytes and Fewer Signs of Astrogliosis

After the colocalization of NLRP3 with the different cell types of the spinal cord, we observed that the most significant increase 24 h and 7 days post-SCI was observed in astrocytes (Figure 3B). Therefore, double immunofluorescence staining of anti-GFAP with anti-NLRP3, anti-PYCARD, and anti-GSDMD was performed at 7 days post-SCI. GFAP showed a colocalization with all NLRP3 inflammasome components in the lesion site of *Lcn2^−/−^* and WT mice (Figure 6). The findings in the lesion area indicated that all inflammasome components including NLRP3 (*p* = 0.0230, *df* = 2, *F* = 14.80), PYCARD (*p* = 0.0313, *df* = 2, *F* = 12.43), and GSDMD (*p* = 0.0412, *df* = 2, *F* = 9.176) were significantly reduced in *Lcn2^−/−^* mice compared with WT mice (Figure 6). On the other hand, only the inflammasome’s downstream molecule GSDMD was significantly decreased (*p* = 0.0494, *df* = 2, *F* = 9.947) in astrocytes, where we observed a lower number of GFAP^+^GSDMD^+^ cells in the lesion site at 7 days post-SCI in *Lcn2^−/−^* mice compared with WT mice (Figure 6). Interestingly, *Lcn2^−/−^* mice showed a decreased GFAP^+^NLRP3^+^ (*p* = 0.1089, *df* = 2, *F* = 10.72) and GFAP^+^PYCARD^+^ (*p* = 0.1482, *df* = 2, *F* = 8.213) population in the lesion site at 7 days post-SCI compared with WT mice, which did not reach statistical significance (Figure 6).

## 3. Discussion

This study shows that LCN2 enhances inflammatory responses after SCI by inducing the priming and activation of the inflammasome and its components NLRP3, PYCARD, and CASP1, with the highest levels at 7 days post-SCI. In addition, we demonstrate that SCI upregulates GSDMD, which has been identified as an executive molecule in pyroptosis. Further, we found that *Lcn2* deficiency improves functional recovery and attenuates inflammatory responses after SCI. LCN2 is a secreted protein (neutrophils, hepatocytes, and renal tubular cells mainly secrete LCN2) of the lipocalin family implicated in the development of several diseases, including cancer [10], cardiovascular disease [19], and stroke [20].

In the acute brain injury model, transient middle cerebral artery occlusion in rats, we already found a significant upregulation of LCN2 in astrocytes 24 h after reperfusion [21]. In previous studies using an SCI contusion model in mice, we also observed that LCN2 protein was increased in the epicenter of the lesion between 6 and 24 h after SCI onset in astrocytes [17]. Astrocyte-derived LCN2 directly influences neuronal function and mediates its neurotoxic effects by enhancing neuroinflammation [11,22]. Therefore, we could conclude that inflammation plays a decisive role in developing spinal cord injury [23,24]. Inflammatory responses are triggered by the accumulation and activation of inflammatory cells in the injured region, such as macrophages, neutrophils, B- and T-lymphocytes, microglia, and astroglia and subsequently proinflammatory mediators released by these cells [25]. An increase in inflammasome activity in these cell types accompanies this early inflammatory scenario. After SCI, the NRLP3 and NLRP1b inflammasomes contribute to the ongoing damage [4,26].

The priming and activation of NLRP3 have been demonstrated in several studies following moderate spinal cord contusion in rats [6] and mice [27]. An exaggerated result of inflammasome activation after SCI may lead to persistent damage to neurons and neural fiber tracts [28]. Targeting the inflammasome activation, thus, is an ingenious therapeutic approach for SCI. The complex cascade of inflammasome priming offers many potential target sites where inhibitors could intervene, directly or indirectly. Recent investigations have disclosed various inhibitors of the NLRP3 inflammasome pathway, which were validated through in vitro studies and in vivo experiments in animal models of NLRP3-associated disorders. Some of these inhibitors directly target the NLRP3 protein, whereas others tackle other components and products of the inflammasome. Indirect targeting can be mediated by inhibiting the NF-kB signaling pathway [29,30]. For example, direct targeting of microRNA-223 inhibits the translation of the NLRP3 protein [31]. Chemical substances such as OLT1177 or BAY 11-7082 interfere with the pathway in human cells in vitro [32] or in mice models in vivo [27]. Subsequently, the downstream molecules are downregulated. For example, MCC950 can bind to the Walker B motif in the NACHT domain of NLRP3; therefore, the activation of NLRP3 is inhibited by blocking ATP hydrolysis [33,34,35]. Whether direct or indirect, all therapeutic approaches led to similar results of mitigated inflammatory responses and improved the functional behavior of the injured animals recently summarized by Das and coworkers [36].

A crucial step in pyroptosis and the pathological progression of SCI is the activation of the NLRP3 inflammasome. Therapies that target NLRP3 are developing quickly, despite our knowledge of the processes behind NLRP3 inflammasome priming, activation, and post-translational modification expanding. The inhibition of inflammasomes is complicated since it involves several membrane and intracellular receptors, numerous signaling pathways and proteins, and transcriptional and post-transcriptional levels. However, the following perspectives seem promising for lowering inflammasome assembly and activation: (1) anti-inflammatory medications that target NLRs (the therapeutic neutralization of NLRP1); (2) ASC-targeting antibodies; (3) targeting proinflammatory cytokines, especially IL-1; and (4) targeting pyroptosis by using antagonists of caspase-1 and pannexin1. Overall, an additional study is still required to fully understand the role of the NLRP3 inflammasome in CNS damage, although innovative therapeutics are rapidly developing and becoming available.

As previously described, we can also report an improved motor function in the current study [17]. The number of studies that have investigated the relationship between LCN2 and SCI is limited. A recent study only focused on the proinflammatory cytokine profile or signaling pathways, such as the JAK–STAT3 pathway, in relation to LCN2 and SCI [37]. The novel aspect of this study is to examine the link between NLRP3 activation, one of the major inflammatory cytokines that control neural inflammation, and LCN2 in the context of SCI. We showed that the deletion of endogenous LCN2 reduced the activation of the NLRP3 inflammasome components HMGB1, NLRP3, PYCARD, CASP1, GSDMD-NT, and IL-1b, which supports the reduction of neuroinflammation. Because no direct interaction between LCN2 and the NLRP3 inflammasome was previously reported in the bioinformatic network analysis of this study, there are different concepts of how LCN2 can regulate the inflammasome activation indirectly. 

A possible interplay may be mediated via the NF-κB pathway, a known transcription factor activating the inflammatory response. In addition, it can be activated by DAMPs, which bind to the TLR4 receptor. The molecule HMGB1 was shown to act as a DAMP in this regard. LCN2 was able to increase the release of HMGB1 in WT mice, while less HMGB1 was found in *Lcn2^−/−^* mice in a study of heart pressure load [38]. In our experimental model, we also confirmed this effect. Accordingly, we observed that *Lcn2^−/−^* mice had lower levels of intracellular HMGB1 compared with WT. Further studies on inflammasome activation regarding HMGB1 are needed to clarify a correlation. 

Another way that LCN2 and the inflammasome interact is through the siderophores it binds, which controls the amount of intracellular iron. According to research, LCN2 can induce cell death in astrocytes and microglia by lowering intracellular iron levels. LCN2 also carries iron into cells, which may help explain why certain Parkinson’s disease patients have an excess of iron in their substantia nigra [22,39,40]. Reactive oxygen species have also been found to have a role in activating the NLRP3 inflammasome by cellular labile iron [41]. Iron may, therefore, also play a mediating role in regulating NLRP3 inflammasome levels.

GSDMD may serve as an additional therapeutic target for NLRP3-induced pyroptosis-related illnesses. CASP1 activates GSDMD, which causes cell swelling and pyroptosis and exacerbates the inflammatory response [42]. In the absence of LCN2, we observed less GSDMD activation, which additionally influenced the inflammasome level due to reduced secretion of inflammatory factors. In a prior investigation of a gastroenteritis norovirus infection, elevated amounts of fecal LCN2 were discovered in Stat-deficient mice, demonstrating the relationship between LCN2 and GSDMD. Interestingly, the infection-related increase of LCN2 was diminished in mice that displayed additional *Nlrp3* or *Gsdmd* deficiency [43]. Therefore, we assume that the release of LCN2 may be a consequence of inflammatory processes and functions as a modulator in a proinflammatory way.

Besides GSDMD, CASP1 also activates IL-1β. We observed a significant reduction in both IL-1β and CASP1 in *Lcn2*^−/−^ mice compared with WT. Therefore we noticed an incongruence between the secretion of IL-1β and gene and protein level and CASP1 activation. Besides the described pathways, NLRP3-independent ways exist that could contribute to this difference [44].

In the past decade, much progress has been made in determining the structure of the NLRP3 inflammasome, its activation mechanisms, and its contribution to the progression of SCI. Furthermore, the association between LCN2 and the inflammasome during SCI supports the possibility that LCN2 interferes with the inflammasome over the NF-κB pathway, leading to increased levels of inflammatory cytokines when present. This makes LCN2 a suitable candidate target to treat and counteract excessive immune responses of the innate immune system.

We also need to emphasize that in this study, we used only male mice, which is a crucial aspect. A number of biological processes may be positively and negatively impacted by sex hormones and how they change with age. Estradiol, for instance, which has been linked to neuroprotective benefits in SCI, dramatically drops after menopause in humans. In investigations on humans and other animals, scientists will take the possible biological relevance of sex into account. Female mice’s hormone regulation changes significantly over the menstrual cycle, in contrast to male sex hormones. Women recuperate from musculoskeletal injuries faster than men, according to clinical and experimental studies. This sex-dependent recovery is primarily attributed to the role of sex hormones in neuroprotection and immunomodulation. LCN2’s impact was evaluated in controlling neuroinflammation without accounting for menstrual variability. Consequently, we are unable to extrapolate the findings from the current study to females. From a pathophysiological point of view, the mechanism of damage in SCI can be divided into primary, due to the trauma itself, and secondary, due to neuroinflammation (which starts just a few hours postinjury and may continue for a long time) and edema. The main treatment goals are to reverse neurological injury, avoid secondary injury, and stabilize the spinal column if necessary. Fast and appropriate interventions affect the clinical outcome, and also identifying the specific mechanisms attributed to secondary injury is critical. Since the present SCI treatment approaches do not effectively initiate functional recovery or axonal regeneration, the concept of neuroinflammation management is accepted worldwide for the reduction of future damages and to provide a better environment for recovery [45]. LCN2 is produced during the innate immune response to bacterial infection as a bacteriostatic factor, and we here demonstrated that interactions between LCN2 and NLRP3 could promote neuroinflammation in the early phase of SCI. Furthermore, several factors in patients with SCI predispose them to different types of severe infectious diseases that could affect morbidity and mortality (even more important than the primary event) and lead to several complications in different organs, which finally increases the cost of management because of prolonged hospitalization [46]. Accordingly, the expression of LCN2 in the central nervous system not only is important in the early stages of SCI, but also may affect the risk of adverse outcome and the quality of life in patients with SCI, especially those who encounter an infection. Therefore, the reduction of LCN2 expression in the brain in the early phase of SCI and the evaluation of its level in patients for effectively tailoring anti-inflammatory treatment and predicting future outcomes would be supportive.

## 4. Material and Methods

### 4.1. Animals and Surgery

For in vivo experiments, male 8- to 14-week-old C57BL/6 and *Lcn2^−/−^* mice (approx. 23 g) were used. The *Lcn2^−/−^* mouse strain was kindly provided by Tak W. Mak, as previously described [47]. The district government’s Review Board Care of Animal Subjects (LANUV, Recklinghausen, North Rhine-Westphalia, Germany, Az 81-02.04.2018.A227) approved the following procedures for animal handling. In cooperation with Tehran, Iran, the *Lcn2^−/−^* mice for 7-day post-SCI experiments were approved by the district government’s Review Board for the Care of Animal Subjects (Ethical No. 962066). WT animals were purchased by Janvier Labs (Le Genest-Saint-Isle, France) and housed at an animal shelter a week before starting surgeries. Four mice per cage were kept on a 12 h light/dark cycle and given water and food ad libitum. The animals were randomly assigned to the experimental groups. Additionally, the surgeon was blinded to the genotype of the mouse. Before surgery, anesthesia was induced with 2–3 vol % isoflurane, kept at 1.5–2 vol % isoflurane during the procedure. Analgesia was achieved by the administration of buprenorphine (0.05–0.1 mg/kg in 100 µL volume, s.c.) 30 min before surgery. Narcosis was controlled by reflex testing (intertoe, lid, and tail tip reflex). The surgery area was disinfected and shaved before the mouse was placed in a stereotactic clamping unit to perform a controlled lesion. Skin incision and blunt dissection of the muscle layers at the area of the T9–T11 level were performed (Figure 7). After laminectomy at T10, the spinal cord was contracted using the Infinite Horizons Spinal Cord Impactor with a force of 60 kdyn (Precision Systems and Instrumentation, Brimstone, LN, USA). Mice in the sham group were subjected only to surgical procedures without SCI. During and after surgery, body temperature was maintained at 36.5 ± 0.5 °C using a heat mat and a rectal probe. Directly after surgery, 0.9% NaCl was injected s.c. to stabilize the blood circulation. Postsurgery, 0.05–0.1 mg/kg buprenorphine was administered s.c. during the day and mixed into the drinking water overnight (1 mg/kg) throughout the experiment. One mouse per cage was housed after surgery, and the urinary bladder was manually emptied twice a day until the mice urinated normally. To prevent infection, the mice received enrofloxacin (Baytril^®^ 2.5%, 0.3 mL/kg s.c., Bayer AG, Leverkusen, Germany; 400614). At the designated time points, animals were finalized, and tissue samples for mRNA, protein, and immunohistochemistry analyses were assigned to a random identification number. The representative fields from the spinal cord that were 0.5 cm distal and 0.5 cm proximal to the center (T10) of the lesion (partial SCI, preservation of sensory and motor function below the level of injury) were prepared and used for molecular (WB and qRT-PCR) and immunofluorescence analysis.

### 4.2. Behavioral Assessment

A modified version of the Basso, Beattie, and Bresnahan (BBB) locomotion rating test for mice, developed by Joshi and Fehlings, was used to evaluate locomotor function. Multiple blind examiners evaluated the BBB scale scores since variations in results across examiners are frequently seen. However, inexperienced observers are able to pick up on the scoring system quickly, and their results soon catch up to those of observers with more experience. The scale monitored open-field mobility for 5 min following SCI, then every day until finalization, measuring body weight support, hind limb movement, forelimb to hind limb coordination, and whole-body motions [48]. 

### 4.3. RNA Extraction and Semiquantitative Real-Time PCR

Total RNA was isolated by phenol–chloroform extraction using peqGold RNA TriFast (PeqLab, Erlangen, Germany; Nr. 30-2020), as previously described [49]. Afterwards, purity and RNA concentration were measured using a NanoDrop 1000 device (PeqLab). For cDNA synthesis, 1 µg of RNA was used, and reverse transcription was performed using an MMLV reverse transcriptase kit and random hexanucleotide primers (Invitrogen, Braunschweig, Germany). Semiquantitative real-time PCR (qRT-PCR) was performed using SYBR Green (Bioline, Braine-Álleud, Belgium, QT615-05) in the CFX Connect qPCR detection system (Bio-Rad, Feldkirchen, Germany). The following protocol was used: 95 °C for 10 min to activate the polymerase, followed by 40 cycles of denaturation for 15 s at 95 °C, primer hybridization for 30 s (temperature as indicated in Table 1), and elongation for 30 s at 72 °C. For evaluation, the relative quantification of the measured mRNA was calculated by the ΔΔCt method using cyclophilin A as the reference gene. All primers used in this study are shown in Table 1.

### 4.4. Tissue Preparation and Immunofluorescence Staining

The spinal cord was removed and postfixed in 3.7% PFA, embedded in paraffin (Merck, Darmstadt, Germany; 8042-47-5) and cut into 5 µm thick slices on the transverse plane. After deparaffinization, a single experienced pathologist, blinded to the experimental groups, stained them with hematoxylin/eosin and luxol fast blue, and documented the SCI grades (0 = absence, 1–3 = presence) according to the extent of pathological outcome, such as hyperemia, degeneration, and infiltration of immune cells. Grade 1 indicated the presence of outcome in up to one-third, grade 2 indicated in one- to two-thirds, and grade 3 indicated in more than two-thirds of the spinal cord width in the lesion area. Finally, all the immunofluorescence staining (Figure 3 and Figure 6) was performed in areas in grade 1 (focusing on white matter lesions in the vicinity of dorsal horns) with three biological replications (max. 50 histological transverse sections per animal).

For double immunofluorescence staining, the spinal cord was cut into 5 µm thick slices. After deparaffinization and heat-induced antigen retrieval, sections were blocked with 2% FCS and 1% BSA in PBS and incubated overnight at 4 °C with the primary antibodies at the given concentration (Table 2). The following day, the sections were incubated with the suitable secondary antibodies for 1 h, including anti-NLRP3, anti-PYCARD, and anti-GSDMD, which were each combined with anti-GFAP to detect astrocytes, anti-NeuN (Rbfox3) antibody to detect neurons, anti-AIF1 to detect microglia, or anti-OLIG2 to detect oligodendrocytes. All sections were counterstained with Hoechst 33342 (Thermo Fisher Scientific, MA, USA; H3570) to visualize the nuclei. Negative controls that gave no signals were acquired by omitting the primary or secondary antibody, respectively. Pictures of the entire region of interest (white and gray matter near the contusion side) were taken in three sections from three mice per group with a Leica DMI6000 B fluorescence microscope (Leica, Wetzlar, Germany). Five areas for each slice were defined in the whole spinal cord for quantifications to ensure similar topography and avoid errors due to the differences in the orientation of planes. Cells were counted in the specified areas of matched planes using ImageJ software (version 1.53; NIH, Bethesda, MD, USA), and all images used for quantification were compared with their respective control tissues. A similar threshold level was set for every image on the dark background, and the positive signals were quantified.

### 4.5. Protein Isolation, SDS-PAGE, and Western Blot

To prepare protein extracts, the central spinal cord was homogenized (Precellys 24, Bertin technologies, Frankfurt am Main, Germany) in RIPA buffer containing protease inhibitors (cOmplete Mini, Roche, Mannheim, Germany; 11836153001). Protein concentrations were determined using a BCA Protein Assay Kit (Pierce, Bonn, Germany; 23225) following the manufacturer’s instructions. For the sodium dodecyl sulfate-polyacrylamide gel electrophoresis (SDS-PAGE), 20 µg of protein was used and separated by 12% (*v*/*v*) gel electrophoresis. Proteins were transferred on PVDF membranes and blocked with 5% milk in Tris-buffered saline (TBS). The membranes were incubated overnight at 4 °C with the respective primary antibody (Table 2)*.* After washing, the membranes were incubated with the appropriate secondary antibody for 2 h at room temperature. For visualization, the enhanced chemiluminescence method (ECL Plus, Pierce Scientific, Waltham, MA, USA) was used according to the manufacturer’s protocol. For evaluation, densitometric quantification was performed. The intensity of the specific bands was measured and normalized to a β-actin or GAPDH as a loading control and quantified using ImageJ (National Institute of Health, Bethesda, MD, USA).

### 4.6. Bioinformatic Analysis

A potential protein–protein interaction network (PPI) between LCN2 and NLRP3 was generated using the STRING database version 11.0 (http://string-db.org, accessed on 10 September 2022) with a confidence value ≥ 0.9 as the threshold [50,51]. In addition, the association of the LCN2 and NLRP3 genes and their variants with human diseases was assessed using DisGeNET version v7.0 (https://www.disgenet.org/, accessed on 20 October 2022) [52,53].

### 4.7. Statistical Analysis

A priori sample size calculation was performed for the animal numbers, gene, and protein studies using G*Power software 3.1 [54]. Statistical tests were calculated using GraphPad Prism 8.3.0 (GraphPad Software Inc., San Diego, CA, USA). The normality of residuals and variance homogeneity were tested using the Brown–Forsythe and Shapiro–Wilk tests. If one of the tests responded significantly, a Box–Cox transformation was performed with optimal λ. A nonparametric test (Kruskal–Wallis) was performed if the data were not distributed normally. Statistical differences between various groups and time points in WT animals and between WT and *Lcn2^−/−^* animals were analyzed with a one-way analysis of variance (ANOVA) and two-way ANOVA, respectively, followed by a Bonferroni post hoc test (multiple comparison test) [55]. All data are given as mean ± standard error of the mean (SEM). Significant differences are indicated as * *p* ≤ 0.05, ** *p* ≤ 0.01, *** *p* ≤ 0.001, and **** *p* ≤ 0.0001 between different experimental groups.

## 5. Conclusions

In this investigation, we showed that the absence of LCN2 suppressed NLRP3 inflammasome formation and reduced neuroinflammatory responses in SCI. This result lends credence to the idea that LCN2 has a proinflammatory role in SCI, influencing the activation of the NLRP3 inflammasome in the second stage after injury. To treat CNS injuries, neutralizing LCN2 with specific antibodies may reduce inflammation and enhance patient outcomes. A prospective study is recommended to assess the role of the gut–brain axis in managing SCI acute inflammation with the central role of LCN2.

## Figures and Tables

**Figure 1 ijms-24-08689-f001:**
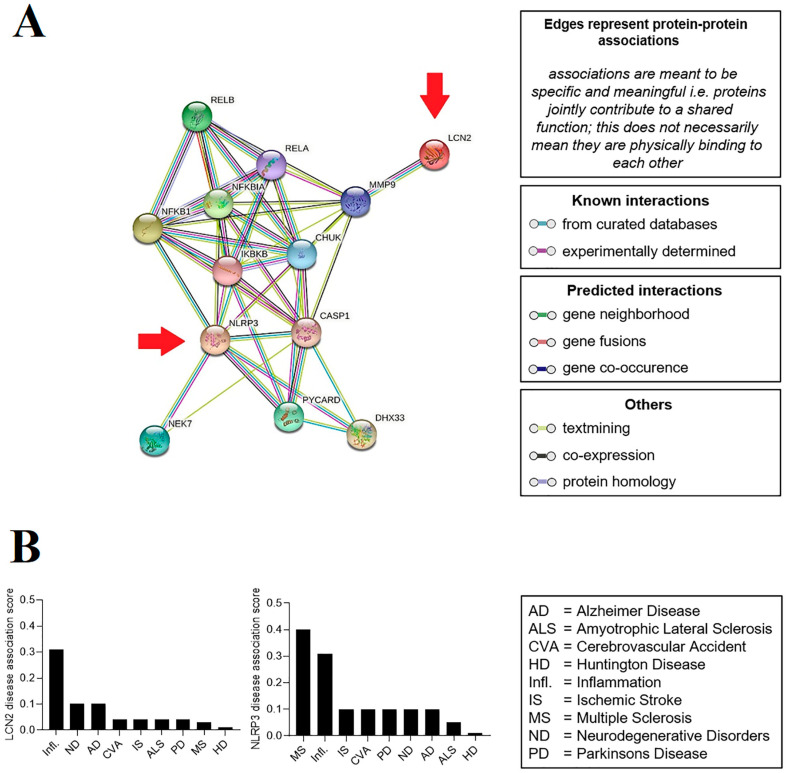
Network analysis and hub gene selection. (**A**) The protein–protein interaction network of the LCN2 and NLRP3 genes (showed by arrows) was created with STRING 11.0, consisting of 13 nodes and 44 edges with high confidence (0.855). Assessment of relationships between target genes and human diseases. (**B**) Possible relationships between the LCN2 and NLRP3 genes and major neuroinflammatory diseases evaluated by DisGeNET using large-scale data analysis.

**Figure 2 ijms-24-08689-f002:**
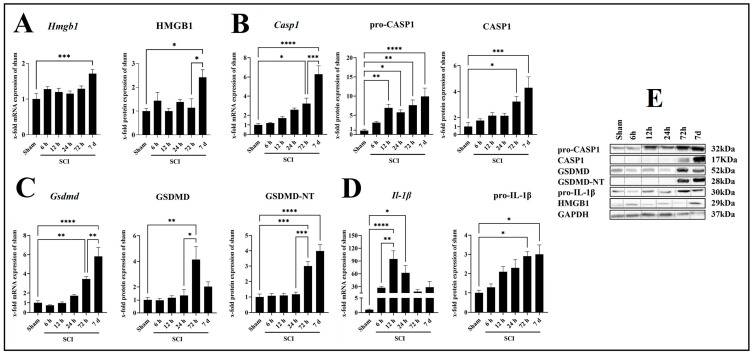
Time course of mRNA and protein level of molecules related to inflammasome activation, *Hmgb1*/HMGB1 (**A**), *Casp1*/pro-CASP1/CASP1 (**B**), *Gsdmd*/GSDMD/GSDMD-NT (**C**), and *Il-1β*/pro-IL-1β (**D**), with corresponding Western blots (**E**). Data are presented as means ± SEM (n = 6 in qRT-PCR and n = 4 in WB groups), analyzed using one-way ANOVA with Bonferroni’s correction for multiple comparisons. * Significances are indicated as * *p* ≤ 0.05, ** *p* ≤ 0.01, *** *p* ≤ 0.001, and **** *p* ≤ 0.0001.

**Figure 3 ijms-24-08689-f003:**
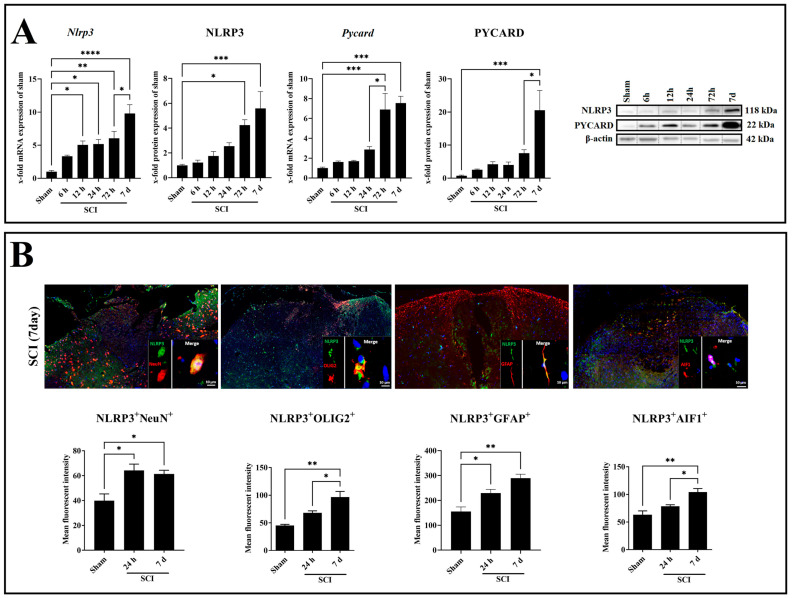
Gene and protein expression of the main inflammasome components NLRP3 and PYCARD in the spinal cord after SCI with corresponding Western blots (**A**). Immunofluorescence evaluation of colocalization of NLRP3^+^/NeuN^+^, NLRP3^+^/GFAP^+^, NLRP3^+^/AIF1^+^, and NLRP3^+^/OLIG2^+^ after spinal cord contusion (**B**). Data are presented as means ± SEM (n = 6 in qRT-PCR, n = 4 in WB, and n = 3 in immunofluorescence groups), analyzed using one-way ANOVA with Bonferroni’s correction for multiple comparisons. Scale bar = 10 µm. * Significances are indicated as * *p* ≤ 0.05, ** *p* ≤ 0.01, *** *p* ≤ 0.001, and **** *p* ≤ 0.0001.

**Figure 4 ijms-24-08689-f004:**
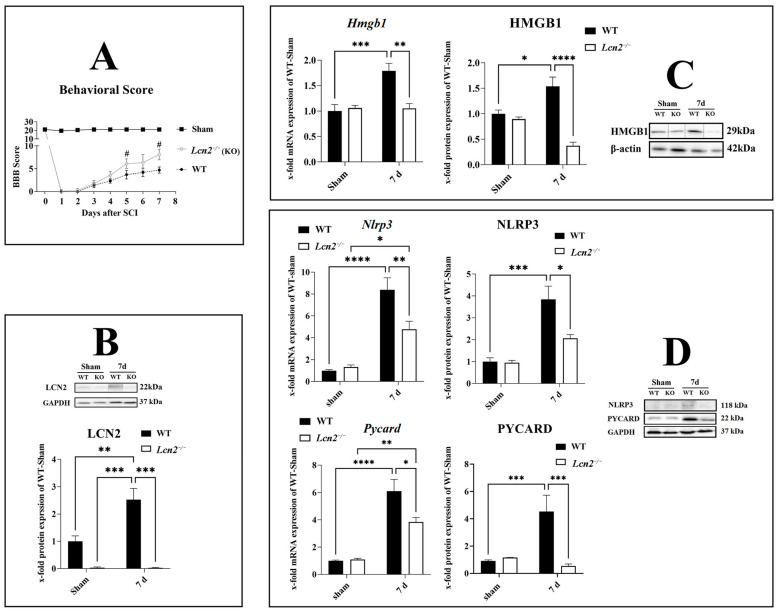
Improvement of behavior score and NLRP3 inflammasome formation in *Lcn2^−/−^* mice. BBB score of WT and *Lcn2^−/−^* mice after different time points after SCI (**A**). LCN2 protein levels were measured in WT and *Lcn2^−/−^* mice 7 days post-SCI (**B**). Gene and protein levels of activated inflammasome component HMGB1 in the spinal cord in WT and *Lcn2^−/−^* mice at 7 days post-SCI with corresponding Western blot (**C**). Gene and protein levels of inflammasome components NLRP3 and PYCARD in the spinal cord in WT and *Lcn2^−/−^* mice at 7 days post-SCI with corresponding Western blots (**D**). Data are presented as means ± SEM (n = 5 in qRT-PCR and n = 5 in WB groups), analyzed using two-way ANOVA with Bonferroni’s correction for multiple comparisons. * Significances are indicated as # (**A**), * *p* ≤ 0.05, ** *p* ≤ 0.01, *** *p* ≤ 0.001, and **** *p* ≤ 0.0001 (**B**–**D**).

**Figure 5 ijms-24-08689-f005:**
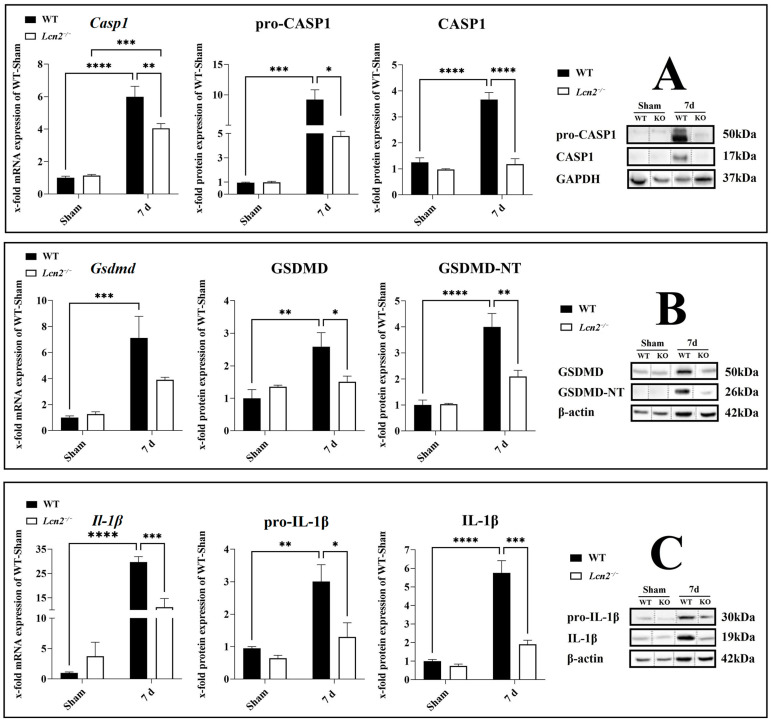
Gene and protein expression of the activated inflammasome components CASP1 (**A**), GSDMD (**B**), and IL-1β (**C**) in the spinal cord after SCI in WT and *Lcn2^−/−^* mice, with corresponding Western blots. Data are presented as means ± SEM (n = 5 in qRT-PCR and n = 5 in WB groups), analyzed using two-way ANOVA with Bonferroni’s correction for multiple comparisons. * Significances are indicated as * *p* ≤ 0.05, ** *p* ≤ 0.01, *** *p* ≤ 0.001, and **** *p* ≤ 0.0001.

**Figure 6 ijms-24-08689-f006:**
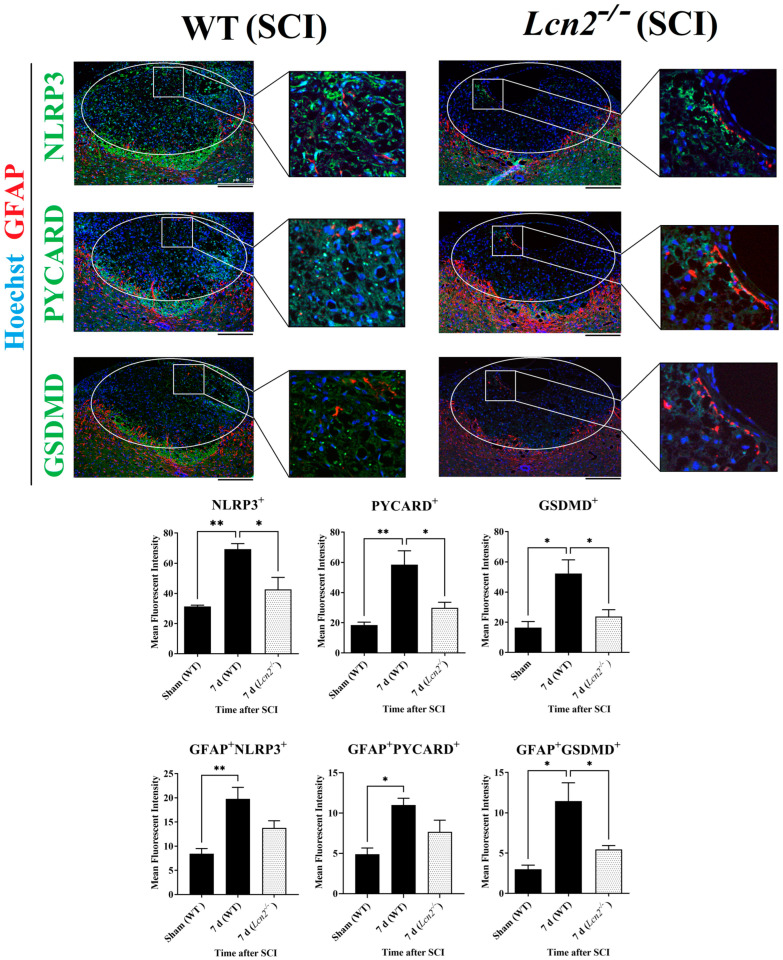
Evaluation of inflammasome formation and colocalization of GFAP with NLRP3, PYCARD, and GSDMD 7 days post-SCI in WT and *Lcn2^−/−^* using immunofluorescence. Data are presented as means ± SEM (n = 3), analyzed using one-way ANOVA with Bonferroni’s correction for multiple comparisons. Scale bar = 250 µm. * Significances are indicated as * *p* ≤ 0.05, ** *p* ≤ 0.01.

**Figure 7 ijms-24-08689-f007:**
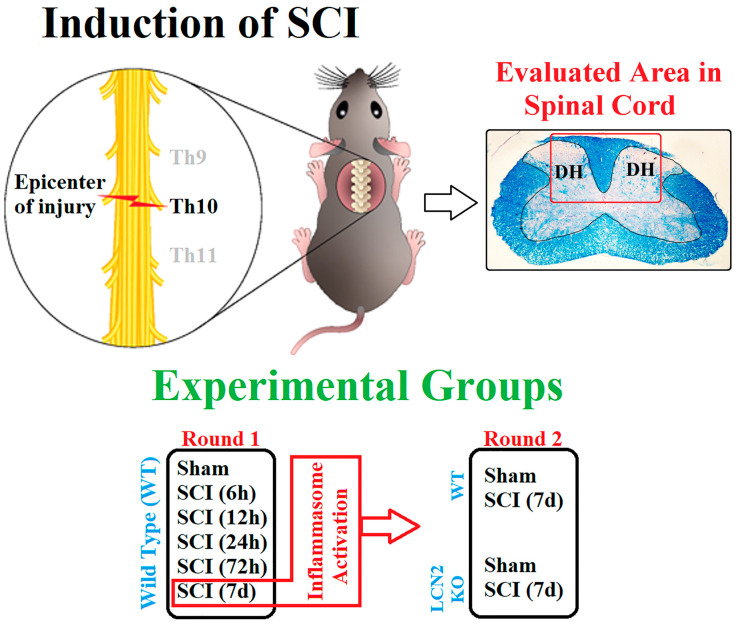
Schematic illustration of SCI and the experimental design. SCI was performed at Th10 on the spinal cord (**up left**). The sample collection was performed approximately 1 cm around this area and damaged white matter and two dorsal horns (DH) on both sides of it targeted for IHC analysis, as shown in the LFB staining section (**up right**). Experiments are designed to have two phases as described (**down**).

**Table 1 ijms-24-08689-t001:** List of used primers with product size and annealing temperature.

Target Genes	Sense	Antisense	Product Size (bp)	AT (°C)
*Gsdmd*	TACTACTACTCGGCTTTCCCGT	TCGAATCTTGTCCAGGGCATC	192	64
*Cyclo A*	TTGGGTCCAGGAATGGCAAGA	ACATTGCGAGCAGATGGGGT	148	62
*Pycard*	CTTGTCAGGGGATGAACTCAAA	GCCATACGACTCCAGATAGTAG	153	60
*Casp1*	CCGTGGAGAGAAACAAGGAGT	CCCCTGACAGGATGTCTCCA	180	62
*Il-1β*	GCCCATCCTCTGTGACTCAT	AGGCCACAGGTATTTTGTCG	230	61
*Nlrp3*	CCTGGGGGACTTTGGAATCAG	GATCCTGACAACACGCGGA	113	65
* Hmgb1 *	GTTACAGAGCGGAGAGAGTG	CCGCAGTTTCCTATCGCTTTG	130	64

**Table 2 ijms-24-08689-t002:** List of used antibodies with concentrations.

Target Protein	Company	Cat. No.	Host	Type	WB	IF
PYCARD	Santa Cruz, Santa Cruz, CA, USA	sc-271054	mouse	mono	1:1000	-
CASP1	Santa Cruz, Santa Cruz, CA, USA	sc56036	mouse	mono	1:400	-
NLRP3/cryopyrin	Bioss, Woburn, MA, USA	bs-10021R	rabbit	poly	1:1000	1:300
GSDMD	Abcam, Cambridge, UK	ab219800	rabbit	poly	1:1000	1:100
HMGB1	Invitrogen, Braunschweig, Germany	ab-2248274	rabbit	Poly	1:1000	-
AIF1	Millipore, Burlington, MA, USA	MABN92	mouse	mono	-	1:500
NeuN	Millipore, Burlington, MA, USA	MAB377	mouse	mono	-	1:2000
GFAP	Santa Cruz, Santa Cruz, CA, USA	sc-33673	mouse	mono	-	1:1000
OLIG2	Millipore, Burlington, MA, USA	MABN50	mouse	mono	-	1:1000
GAPDH	Santa Cruz, Santa Cruz, CA, USA	sc-25778	rabbit	poly	1:5000	-
β-actin	Santa Cruz, Santa Cruz, CA, USA	sc-47778	mouse	mono	1:5000	-
IL-1β	Cell Signaling, Danvers, MA, USA	63124S	rabbit	mono	1:1000	
LCN2	Antibodies Online, Philadelphia, PA, USA	ABIN107840	rabbit	poly	1:500	-

WB = Western blot; IF = immunofluorescence.

## Data Availability

Datasets generated and analyzed during this study are available from the corresponding author upon reasonable request.

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
