# Peer review of "Lipocalin-2 Deficiency Diminishes Canonical NLRP3 Inflammasome Formation and IL-1β Production in the Subacute Phase of Spinal Cord Injury"

_ijms, 2023, doi:10.3390/ijms24108689_

Round 1
Reviewer 1 Report
The manuscript is very well written. Experiments are well designed and properly interpreted. The results are also sufficiently discussed.
I recommend acceptance of the manuscript.
The references need to be put in order.
Fig. 3A blot needs to be labelled or put in the same box as 3A.
Author Response
The manuscript is very well written. Experiments are well designed and properly interpreted. The results are also sufficiently discussed.
I recommend acceptance of the manuscript.
- The references need to be put in order.
We appreciate this valuable suggestion. We corrected the references in the new version of the manuscript accordingly.
- 3A blot needs to be labelled or put in the same box as 3A.
We appreciate this valuable suggestion. We corrected it in the new version of the manuscript accordingly.
Reviewer 2 Report
In order to improve the manuscript, several changes must be introduced in the final version:
INTRODUCTION
The information that the authors provide in the introduction of the manuscript is very interesting, however its translation to the clinic or preclinical is not justified. Knowing the interaction between LCN2 and NLRP3 which functional results can improve in subjects with spinal cord injury? Could this molecular knowledge improve motor recovery in subjects with spinal cord injury, since it is a key signaling pathway in the survival of spinal motor neurons after traumatic spinal cord injury? Does it decrease central neuropathic pain? Does it reduce syringiomyelia in subjects with spinal cord injury? The authors must provide key information on the need to continue investigating this intracellular signaling pathway in the context of real clinical and preclinical translation to improve the life of the subject with spinal cord injury. Include this highly relevant information in the final version of the manuscript.
Will the information derived from the investigation of this manuscript, which aims to elucidate the molecular interactions between LCN2 and NLRP3, allow the short-term development of real pharmacological treatments that improve the quality of life of patients with spinal cord injury? Why is it very important to focus financial and human resources on investigating this signaling pathway in subjects with spinal cord injury? Please answer these questions and include this relevant information in the final version of the manuscript.
RESULTS
Figure 4 shows histological images of double labeling. What area of the injured spinal cord were these images taken from? White matter or gray matter? Injury area or perilesional area? Please include this information in the final version of the manuscript.
In figure 4 it is assumed that the histological images correspond to confocal microscope images. If so, the use of this microscope should be included in the methodology. If not, I suggest indicating how these images were obtained.
Figure 7 shows histological images of double labeling. Are the spinal cord areas analyzed between the WT group and the Lcn2-/- group equivalent? The images in figure 7 correspond to which region of the injured spinal cord? Please include all this information in the final version of the manuscript.
In figure 7 it is difficult to identify the decrease in astrogliosis indicated by the authors in the text. More images should be provided where this reduction in astrogliosis can be better observed.
In Figure 7, the calibration bars are missing from all histology images.
DISCUSSION
The degree of motor functional recovery observed in this study is very low, since on a 21-point scale, KO animals present values below 10 points. It should be noted that spinal cord contusion is a model of incomplete injury of the spinal cord, in which, with a minimal edge of spinal cord preserved in the ventral area, the animals already have the ability to move. In this context, the authors' discussion is poor and poorly contextualized with previous studies of concussion spinal cord injury. Please include relevant information on spinal cord injury and contextualize the discussion initially in the experimental model used. Include all this relevant information in the final version of the manuscript.
The study of LCN2 in the context of spinal cord injury is not new, there are already previous studies that have analyzed this point. The authors do not discuss these previous studies in depth in relation to the results obtained in the manuscript. What is the novelty of the results of the manuscript with respect to previous studies? Please include all this new information in the final version of the manuscript.
In the scientific literature there are scientific articles prior to this manuscript that assess the relationship between NLRP3 and spinal cord injury. The authors should thoroughly review this bibliography and put all these previous articles in context in relation to the results obtained in the manuscript, highlighting the novelty of this manuscript compared to the previous ones.
What is the real scientific progress of the research carried out by the authors in the context of spinal cord injury and the degree of functional recovery (motor, sensory, emotional, etc.) of subjects suffering from spinal cord injury? What is the new relevant contribution of the research described in this manuscript that allows the development of new therapeutic approaches that improve the quality of life of spinal cord injuries? Authors should clarify and discuss these points in depth. This information is very relevant and should be included in the final version of the manuscript.
The authors when writing the manuscript have used a template of the year 2021 (Int. J. Mol. Sci. 2021, 22, x. https://doi.org/10.3390/xxxxx). Please use a more current template from the year 2023.
MATERIALS AND METHODS
Why have the authors used male animals? Why have not female animals also been used? Please clarify this point and include this relevant information in the final version of the manuscript.
The authors have performed a contusion injury to the spinal cord. What is the criteria for using an impact force of 60 kdyn? Please clarify this point and include this information in the final version of the manuscript.
The spinal cord contusion was performed at T10 level. Considering that secondary spinal cord injury spans at least 1 to 2 spinal cord segments from the primary area of injury, the T12-L1 spinal cord segments will most likely be involved. These medullary levels contain the locomotion spinal control center, which may be affected by the lesion induced by the authors choosing T10 as the primary zone of lesion. What was the criteria for performing spinal cord contusion at T10? Please clarify this point and include this relevant information in the final version of the manuscript.
The authors assess the degree of motor recovery using the open field locomotion test and the BBB scale. The authors should describe this test more and better. Likewise, the BBB scale is an international rat scale, while the BMS scale is an international mouse scale. Taking into account that the authors have used mice, why do they use the BBB scale and not the BMS scale? Clarify these points and include this information in the final version of the manuscript.
For molecular and histological assessments, which segment of the spinal cord was used? This information must be included in the final version of the manuscript.
In this methodology section, the authors indicate the existence of tables (1 and 2), which have not been included in the version sent to the reviewers. Please, include these tables or include relevant information about the commercial houses where all the chemical products used in the experiment described in the manuscript have been purchased.
How many histological sections per animal have been analyzed? Are these sections longitudinal or transverse? Please include this information in the final version of the manuscript.
Author Response
In order to improve the manuscript, several changes must be introduced in the final version:
INTRODUCTION
- The information that the authors provide in the introduction of the manuscript is very interesting, however its translation to the clinic or preclinical is not justified. Knowing the interaction between LCN2 and NLRP3 which functional results can improve in subjects with spinal cord injury? Could this molecular knowledge improve motor recovery in subjects with spinal cord injury, since it is a key signaling pathway in the survival of spinal motor neurons after traumatic spinal cord injury? Does it decrease central neuropathic pain? Does it reduce syringiomyelia in subjects with spinal cord injury? The authors must provide key information on the need to continue investigating this intracellular signaling pathway in the context of real clinical and preclinical translation to improve the life of the subject with spinal cord injury. Include this highly relevant information in the final version of the manuscript.
It is becoming clearer how Lcn2 acts in the innate immune system's protection against pathogens and infections. Its involvement in other inflammatory diseases and severe traumas, however, has not been fully investigated. We present evidence that Lcn2 is expressed after SCI and that Lcn2 has a negative impact on SCI by causing the loss of neurons and astrocytes, the release of pro-inflammatory cytokines, and the invasion of immune cells. In another study, we showed that NLRP3 inflammasome complexes and their modules were successively activated in the zone of injury to tissue following SCI. We will investigate if LCN2 influences the activation of NLRP3 because both molecules are involved in processing neuroinflammation in the pathophysiology of SCI.
- Will the information derived from the investigation of this manuscript, which aims to elucidate the molecular interactions between LCN2 and NLRP3, allow the short-term development of real pharmacological treatments that improve the quality of life of patients with spinal cord injury? Why is it very important to focus financial and human resources on investigating this signaling pathway in subjects with spinal cord injury? Please answer these questions and include this relevant information in the final version of the manuscript.
We appreciate this valuable suggestion. We have now included the paragraph describing this concern at the end of the discussion (Lines 348 to 351).
RESULTS
- Figure 4 shows histological images of double labeling. What area of the injured spinal cord were these images taken from? White matter or gray matter? Injury area or perilesional area? Please include this information in the final version of the manuscript.
We appreciate this valuable suggestion. We have included the paragraph describing this concern at the material and method (Lines 299 to 318).
- In figure 4 it is assumed that the histological images correspond to confocal microscope images. If so, the use of this microscope should be included in the methodology. If not, I suggest indicating how these images were obtained.
We apologize if the explanation was confusing. The immunohistochemistry staining has been photographed using a standard florescent microscope not a confocal one.
- Figure 7 shows histological images of double labeling. Are the spinal cord areas analyzed between the WT group and the Lcn2-/- group equivalent? The images in Figure 7 correspond to which region of the injured spinal cord? Please include all this information in the final version of the manuscript.
We appreciate this valuable suggestion. We have now included the paragraph describing this concern at the material and method (Lines 374 to 381).
- In figure 7 it is difficult to identify the decrease in astrogliosis indicated by the authors in the text. More images should be provided where this reduction in astrogliosis can be better observed.
We appreciate this valuable suggestion. We corrected this sentence in the text. We don’t evaluate only the GFAP population which enables us to discuss astrogliosis alone. With double staining for GFAP and NLRP3, we can only talk about the astrocytes population with active inflammasome complex.
- In Figure 7, the calibration bars are missing from all histology images.
We added the scale bars for all histology images.
DISCUSSION
- The degree of motor functional recovery observed in this study is very low, since on a 21-point scale, KO animals present values below 10 points. It should be noted that spinal cord contusion is a model of incomplete injury of the spinal cord, in which, with a minimal edge of spinal cord preserved in the ventral area, the animals already have the ability to move. In this context, the authors' discussion is poor and poorly contextualized with previous studies of concussion spinal cord injury. Please include relevant information on spinal cord injury and contextualize the discussion initially in the experimental model used. Include all this relevant information in the final version of the manuscript.
In this study, the Infinite Horizon impactor device was used to mimic the pathology of spinal cord injury in mice, using a force-controlled impact to inflict injury. A stepper motor interfaces with an external computer to deliver a controlled impact. After laminectomy, a metal impactor inflicts injury and an attached sensor directly measures the force between the impactor and the spinal cord tissue. This minimises the error introduced by specimen movement. When the pre-set force threshold is reached, the tip is automatically and immediately withdrawn and the exact force transmitted is displayed. This eliminates the weight bounce phenomenon. Three different severities of injury mimic the three types of injury: mild, moderate and severe. In this study, SCI was induced at 60 kdyn, mimicking the moderate SCI model. This allows the animals to have functional motor improvement during the week. However, animals with a BBB score greater than 0 at 24 hours post injury were excluded from the study.
- The study of LCN2 in the context of spinal cord injury is not new, there are already previous studies that have analyzed this point. The authors do not discuss these previous studies in depth in relation to the results obtained in the manuscript. What is the novelty of the results of the manuscript with respect to previous studies? Please include all this new information in the final version of the manuscript.
We appreciate this valuable suggestion. We have now included the paragraph describing this concern at the material and method (Lines 299 to 318).
- In the scientific literature there are scientific articles prior to this manuscript that assess the relationship between NLRP3 and spinal cord injury. The authors should thoroughly review this bibliography and put all these previous articles in context in relation to the results obtained in the manuscript, highlighting the novelty of this manuscript compared to the previous ones.
Many thanks for your input; the discussion has been modified in considering it.
- What is the real scientific progress of the research carried out by the authors in the context of spinal cord injury and the degree of functional recovery (motor, sensory, emotional, etc.) of subjects suffering from spinal cord injury? What is the new relevant contribution of the research described in this manuscript that allows the development of new therapeutic approaches that improve the quality of life of spinal cord injuries? Authors should clarify and discuss these points in depth. This information is very relevant and should be included in the final version of the manuscript.
We appreciate this valuable suggestion. We have now included the paragraph describing this concern at the end of the discussion (Lines 348 to 351).
- The authors when writing the manuscript have used a template of the year 2021 (Int. J. Mol. Sci. 2021, 22, x. https://doi.org/10.3390/xxxxx). Please use a more current template from the year 2023.
I appreciate your input. We implemented the required corrections in the paper.
MATERIALS AND METHODS
- Why have the authors used male animals? Why have not female animals also been used? Please clarify this point and include this relevant information in the final version of the manuscript.
The impact of sex hormones and how they fluctuate with age may have a positive and negative impact on a variety of biological processes. For instance, oestradiol, which is known to have neuroprotective effects in SCI, drastically decreases in human beings following menopause. Researchers will consider the potential biological significance of sex in studies on humans and vertebrates. Contrary to male sex hormones, female mice's hormone regulation modifications significantly during the menstrual cycle. Clinical and experimental studies have shown that females heal more quickly from musculoskeletal injury. The function of sex hormones in neuroprotection and immunomodulation is largely responsible for this sex-dependent recovery. Following traumatic brain injury, sex differences in acute inflammation have been discovered. The current neuroprotective impact of female sex hormones has been reduced by employing male mice in the current study since it addresses acute neuroinflammation following SCI. So that we get to assess LCN2's influence on regulating neuroinflammation without taking into account menstrual fluctuation
- The authors have performed a contusion injury to the spinal cord. What is the criteria for using an impact force of 60 kdyn? Please clarify this point and include this information in the final version of the manuscript.
and
- The spinal cord contusion was performed at T10 level. Considering that secondary spinal cord injury spans at least 1 to 2 spinal cord segments from the primary area of injury, the T12-L1 spinal cord segments will most likely be involved. These medullary levels contain the locomotion spinal control center, which may be affected by the lesion induced by the authors choosing T10 as the primary zone of lesion. What was the criteria for performing spinal cord contusion at T10? Please clarify this point and include this relevant information in the final version of the manuscript.
Although thoracic SCIs are the most frequently used location in animal models, experimental studies show that cervical SCIs are the most frequently involved region in SCIs in humans (Orr and Gensel 2018). Thoracic SCI models appear to be accurate and simple to replicate based on the literature (Soblosky, Song et al. 2001). SCIs at the thoracic levels allow the isolation and analysis of white matter losses since the loss of gray matter in this area of the spinal column results in less obvious functional loss (Lane, Fuller et al. 2008). T12 was chosen for SCI in this investigation because we intended to target the lower thoracic spinal cord and, to the greatest extent efficient, avoid interfering with the sympathetic innervation of other organs like liver. In this study, SCI was induced at 60 kdyn, mimicking the moderate SCI model. This allows the animals to have functional motor improvement during the week. However, animals with a BBB score greater than 0 at 24 hours post injury were excluded from the study.
- The authors assess the degree of motor recovery using the open field locomotion test and the BBB scale. The authors should describe this test more and better. Likewise, the BBB scale is an international rat scale, while the BMS scale is an international mouse scale. Taking into account that the authors have used mice, why do they use the BBB scale and not the BMS scale? Clarify these points and include this information in the final version of the manuscript.
Thanks for this comment. The BBB Locomotor Rating Scale is a reliable and reproducible tool, which involves assessment of the movement of hip joints, forelimb–hindlimb coordination, and toe clearance in the recovery phase after SCI (Basso, Beattie, & Bresnahan, 1995). The original BBB model has been used frequently in SCI experiments in rats which was developed for the assessment of open-field walking. Since other established locomotor tests for spinal cord injured animals, such as the Tarlov scale and the inclined-plane test, would not be adequate for application in SCI-treated mice, (because it is very difficult to obtain accurate measurements in restless mice), the BBB scale was also applied to the assessment of functional outcomes in spinal cord injured mice. Since mouse recovery differed from rats for coordination, paw position and trunk instability and also due to their rapid movement and smaller size, in our study we have used the modified version of test for mice which was designed by Joshi and Fehlings. The BBB Scale scores should be assessed by multiple blind examiners because differences in scores among examiners are often seen. However, inexperienced observers are able to quickly learn the scoring method and their scores quickly approach those of experienced observers.
- For molecular and histological assessments, which segment of the spinal cord was used? This information must be included in the final version of the manuscript.
We appreciate this valuable suggestion. We have included the paragraph describing this concern at the material and method (Lines 299 to 318, Fig. 1).
- In this methodology section, the authors indicate the existence of tables (1 and 2), which have not been included in the version sent to the reviewers. Please, include these tables or include relevant information about the commercial houses where all the chemical products used in the experiment described in the manuscript have been purchased.
We added these tables to the submitted materials.
- How many histological sections per animal have been analyzed? Are these sections longitudinal or transverse? Please include this information in the final version of the manuscript.
We appreciate this valuable suggestion. We have included the paragraph describing this concern at the material and method (Lines 299 to 318, Fig. 1).
Reviewer 3 Report
Manuscript ID: ijms-2283627
Manuscript Title: Anti-microbial peptide lipocalin-2 deficiency diminishes canonical NLRP3 inflammasome formation and IL-1β production in the subacute phase of spinal cord injury
Authors investigated the effects of LCN2 against SCI damage in the spinal cord based on HMGB1/NLRP3/PYCARD/ Caspase-1 axis, IL-1β production. In addition, they confirmed the results in Lcn2-/- mice. The results are interesting, but there are some concerns to consider for publication.
In Figure 4, authors demonstrated the colocalization of NLRP3/neuN, NLRP3/GFAP, NLRP3/AIF1, and NLRP3/OLIG2 after spinal cord contusion. I suggest that authors should demonstrate low magnified pictures as well as 24 h or 7 days after SCI in Figure 4.
In Figure 7, authors demonstrated the colocalization of GFAP with NLRP3, PYCARD, and GSDMD after spinal cord contusion after 7 days in WT and Lcn2-/- . Authors described that they observed that the most significant increase (in shorter time point) was observed in astrocytes post-SCI after co-localization of NLRP3 with the different cell types of the spinal cord. However, the ratio of increases in mean fluorescent intensity was similar in GFAP, AIF1, and OLIG2.
Why did the authors focus on astrocytes, not both astrocytes and microglia? I suggest that authors should conduct the double immunofluorescent staining for AIF1 and NLRP3, PYCARD, and GSDMD.
Author Response
Authors investigated the effects of LCN2 against SCI damage in the spinal cord based on HMGB1/NLRP3/PYCARD/ Caspase-1 axis, IL-1β production. In addition, they confirmed the results in Lcn2-/- mice. The results are interesting, but there are some concerns to consider for publication.
- In Figure 4, authors demonstrated the colocalization of NLRP3/neuN, NLRP3/GFAP, NLRP3/AIF1, and NLRP3/OLIG2 after spinal cord contusion. I suggest that authors should demonstrate low magnified pictures as well as 24 h or 7 days after SCI in Figure 4.
We appreciate this valuable suggestion. We updated this image according to your suggestion.
- In Figure 7, authors demonstrated the colocalization of GFAP with NLRP3, PYCARD, and GSDMD after spinal cord contusion after 7 days in WT and Lcn2-/- . Authors described that they observed that the most significant increase (in shorter time point) was observed in astrocytes post-SCI after co-localization of NLRP3 with the different cell types of the spinal cord. However, the ratio of increases in mean fluorescent intensity was similar in GFAP, AIF1, and OLIG2.
We appreciate this valuable suggestion. The aim of the first evaluation (Fig. 4) was to find the best time point and suitable glial cell for future analysis. As you can see in this figure after 24h only NeuN and GFAP markers increased significantly and thereafter only GFAP also showed more increase at 7d continuously. Moreover, the expression of Olig2 and AIF1 significantly increased only at 7d. With these preliminary results, we concluded that astrocytes are the glial cells that earlier than other cells affected or sensed the SCI damage and overexpressed NLRP3 and selected these cells for future studies in knock-out animals. So, the later (7d) activation of NLRP3 in other glial cells could be mediated by the release of inflammatory factors by astrocytes. We can not confirm this hypothesis by our current results but ablation of astrocytes could be recommended for future complimentary studies. So, to answer your question, the ratio of increases in mean fluorescent intensity was similar in GFAP, AIF1, and OLIG2 but in 7d and not in 24h.
- Why did the authors focus on astrocytes, not both astrocytes and microglia? I suggest that authors should conduct the double immunofluorescent staining for AIF1 and NLRP3, PYCARD, and GSDMD.
As, we mentioned before, from our preliminary results (Fig. 4), we concluded that astrocytes are the glial cells that earlier than other cells affected or sensed the SCI damage and overexpressed NLRP3 and selected these cells for future studies in knock-out animals. Anyway, double immunofluorescent staining for AIF1 and NLRP3, PYCARD, and GSDMD could be also recommended for future complimentary studies.
Round 2
Reviewer 2 Report
It is appreciated that the authors have made an effort to respond to all the questions and suggestions made by the reviewer, and that they have incorporated most of the information in the final version of the manuscript. This has significantly improved the quality of this manuscript.